# The consequences of exercise-induced weight loss on food reinforcement. A randomized controlled trial

Kyle D. Flack [ID]*, Harry M. Hays[☯], Jack Moreland[☯]

Department of Dietetics and Human Nutrition, University of Kentucky, Lexington, Kentucky, United States of America

☯ These authors contributed equally to this work.

* Kyle.Flack@uky.edu

## Abstract

### Background

Obesity remains a primary threat to the health of most Americans, with over 66% considered overweight or obese with a body mass index (BMI) of 25 kg/m$^2$ or greater. A common treatment option many believe to be effective, and therefore turn to, is exercise. However, the amount of weight loss from exercise training is often disappointingly less than expected with greater amounts of exercise not always promoting greater weight loss. Increases in energy intake have been prescribed as the primary reason for this lack of weight loss success with exercise. Research has mostly focused on alterations in hormonal mediators of appetite (e.g.: ghrelin, peptide YY, GLP-1, pancreatic polypeptide, and leptin) that may increase hunger and/or reduce satiety to promote greater energy intake with exercise training. A less understood mechanism that may be working to increase energy intake with exercise is reward-driven feeding, a strong predictor of energy intake and weight status but rarely analyzed in the context of exercise.

### Design

Sedentary men and women (BMI: 25–35 kg/m$^2$, N = 52) were randomized into parallel aerobic exercise training groups partaking in either two or six exercise sessions/week, or sedentary control for 12 weeks.

### Methods

The reinforcing value of food was measured by an operant responding progressive ratio schedule task (the behavioral choice task) to determine how much work participants were willing to perform for access to a healthy food option relative to a less healthy food option before and after the exercise intervention. Body composition and resting energy expenditure were assessed via DXA and indirect calorimetry, respectively, at baseline and post testing.

**Data Availability Statement:** All data is available on figshare, DOI: 10.6084/m9.figshare.12081624.

**Funding:** The project described was supported by the NIH National Center for Advancing Translational Sciences through grant number UL1TR001998.

The content is solely the responsibility of the authors and does not necessarily represent the official views of the NIH. The funders had no role in study design, data collection and analysis, decision to publish, or preparation of the manuscript.

**Competing interests:** The authors have declared that no competing interests exist.

## Results

Changes in fat-free mass predicted the change in total amount of operant responding for food (healthy and unhealthy). There were no correlations between changes in the reinforcing value of one type of food (healthy vs unhealthy) to changes in body composition.

## Conclusion

In support of previous work, reductions in fat-free mass resulting from an aerobic exercise intervention aimed at weight loss plays an important role in energy balance regulation by increasing operant responding for food.

## Introduction

The prevalence of obesity and its comorbidities, including cardiovascular disease, hypertension, diabetes, dyslipidemia, metabolic syndrome, and certain cancers, are still plaguing the nation today [1–4]. Exercise is a long-standing remedy for nearly all of obesity's comorbidities and often recommended as an economical and health-promoting option for weight loss and weight loss maintenance [5]. Unfortunately, as supported by the ever-escalating prevalence of obesity, exercise is often only marginally successfully at reducing body weight [6]. The lack of weight-loss success with exercise has been blamed on a coordinated set of compensatory mechanisms the human body uses to maintain energy balance, thereby resisting the sustained negative energy balance required for weight loss [7]. These compensatory responses may be physiological, and include reductions in resting energy expenditure and metabolic efficiency in attempts to return the body back to energy balance [7]. However, the most prevalent mechanism responsible for maintaining energy homeostasis during an exercise program is increases in energy intake, largely due to the fact that the rate of energy intake far exceeds the rate of energy expenditure [7–9]. For instance, it takes most people between 40 and 60 minutes to expend 500 kcal through exercise, which is about the energy content of a fast-food cheeseburger that can be consumed in 5–10 minutes.

There has been a great deal of research on exercise's effects on energy intake, specifically by assessing how hormonal mediators of appetite, lab-based food intake, and hunger/satiety scales change with exercise [10–14]. However, many have demonstrated single bouts of exercise do not result in changes in appetite, food intake, or appetite-regulating hormones [15–18]. Some have actually demonstrated greater perceptions of hunger and fullness persist 24 hours after exercise [19], and long-term exercise improves the satiety response to a meal [13, 20]. These results are at odds with the apparent compensatory response individuals display, therefore necessitating novel and innovative research to better understand the causes behind exercise-induced increases in energy intake. Empirical hypotheses postulate that individuals who exercise for weight loss have distorted portion control, seek rewards for exercising in the form of food, and derive greater pleasure from high-fat, high-sugar, energy-dense food [7, 21, 22], all of which may be independent of hunger. These behaviors may be explained by a neuro-behavioral cross-talk that is in play with exercise and eating, as both are products of the central dopamine system [23, 24]. There is evidence of such direct cross-talk between eating and exercise reward in rats, whereas a small dose of wheel running that produces a low-level dopaminergic response facilitates eating [25]. A similar neuro-behavioral cross-talk between the reinforcing effects of different drugs is also known, specifically involving endogenous opioid and

cannabinoid systems, whereas administration of one increases the reinforcing value of the other [26]. The rewarding aspects of food and eating are important volitional behavioral responses, assessed by defining one's relative reinforcing value of food ($RRV_{food}$) or food reinforcement [27, 28]. Food reinforcement is a measure of how much an individual wants to engage in a particular eating behavior, as someone who finds food highly reinforcing "wants" food to a greater degree. $RRV_{food}$ is the reinforcing value of food relative to an alternative, which may be a non-eating activity or a different type of food (high vs. low energy density for example). Importantly, food reinforcement is a more robust predictor of food intake than the hedonic value (liking) of the food [28] and is a strong predictor of body weight and energy intake [27, 29]. It therefore seems plausible that the reinforcing value of food, is increased by exercise to promote food intake, although very little research has been done on this issue. Understanding these volitional behavioral responses could inform future weight loss and exercise recommendations by adding important considerations when exercising for weight control. The present study hypothesized that a 12-week aerobic exercise for weight loss intervention would increase food reinforcement, specifically for the high energy-dense foods provided (assessed by an operant responding task) among sedentary, overweight to obese humans. It was further hypothesized that a greater dose of exercise would evoke a greater effect on food reinforcement and that changes in body composition would be correlated with changes in food reinforcement.

## Materials and methods

### Participants

A total of 80 participants aged 18 to 40 years volunteered and were enrolled into the study. Of these, 52 completed all baseline tests and were randomized into one of three groups (six exercise sessions per week, two sessions per week, and sedentary control) during this longitudinal, randomized, controlled trial on a 1:1:1 ratio. Of these 52 randomized participants, 44 completed the study (32 female), with six (four female) withdrawing for personal reasons and two females being excluded for non-compliance (did not complete the required 85% of exercise sessions assigned per month). A consort diagram is depicted in Fig 1. All participants had a body mass index (BMI) ranging from 25–35 kg/m$^2$ and were inactive (not engaging in any form of exercise), determined during screening where participants were asked of their exercise behaviors. Participants were also free of medical issues that would be a contraindication to exercise, without metabolic or cardiovascular disease, not engaging in a weight loss diet, were weight stable (not lost or gained 5% of their current body weight in the previous 6 months), not taking medications that may influence appetite, and not pregnant or nursing. Recruitment began in the winter of 2018 and continued until recruitment goals were met (spring of 2019) in and around Lexington, Kentucky. Participants were a sample who responded to recruitment media including printed brochures and flyers and online advertisements placed on University of Kentucky's Center for Clinical and Translational Science (CCTS) website. This study was approved by the University of Kentucky Institutional Review Board. The present analysis is a secondary outcome of a trial aimed at assessing mechanisms of energy compensation at different doses of exercise, ClinicalTrials.gov identifier: NCT03413826.

### Procedures

During the initial screening and consenting visit, participants provided their written informed consent and were screened of eligibility criteria, completing a physical activity readiness questionnaire (PARQ), health history questionnaire, and screened on their dieting, weight loss history, and physical activity behaviors. Participants also completed a taste-test of the study foods

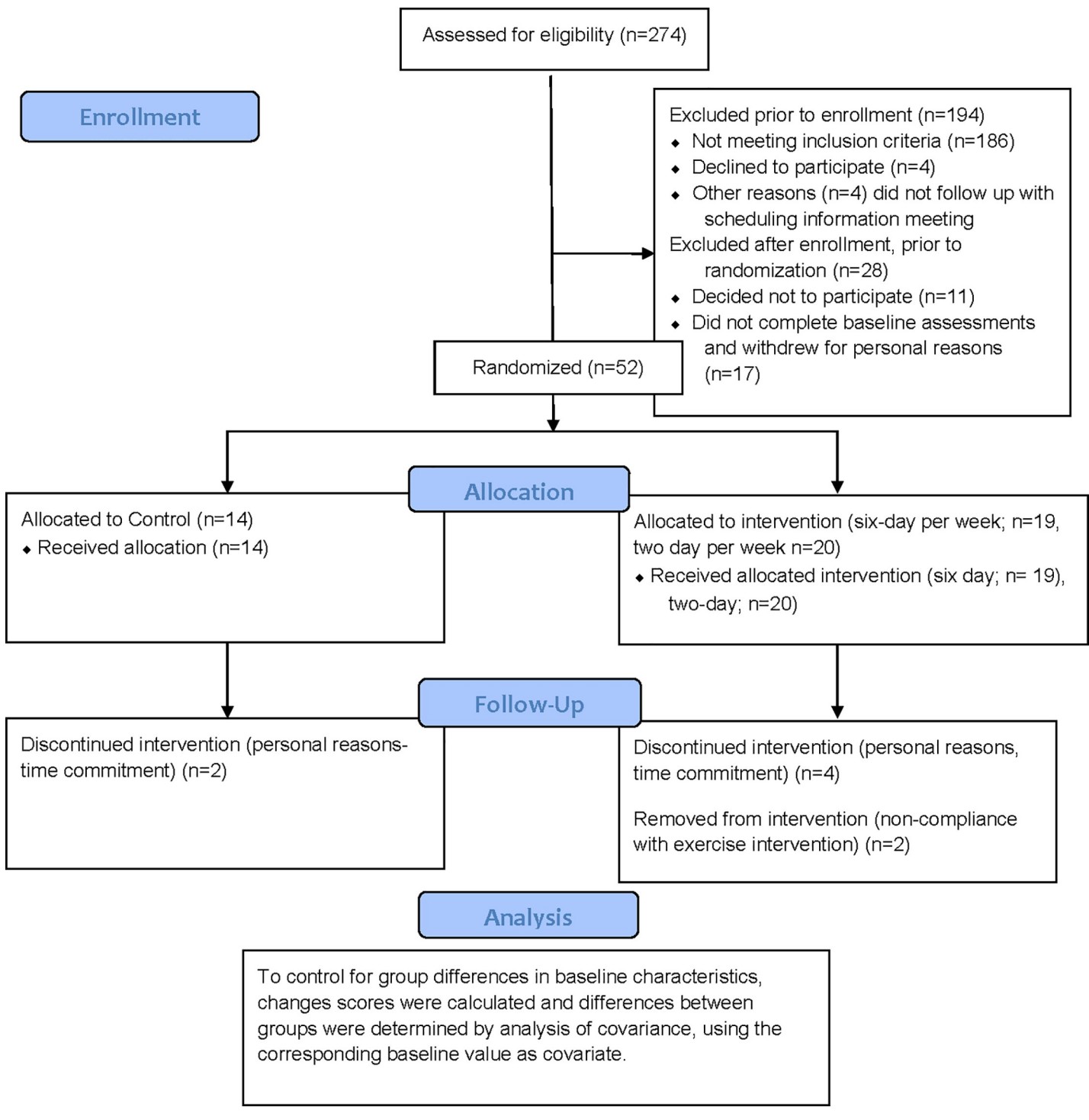

**Fig 1. Consort diagram.** Figure depicting recruitment, retention, and randomization of present trial.

and were scheduled for other assessments including rate of energy expenditure (resting and during exercise), body composition, and food reinforcement (all detailed below).

**Study design.** The study was a randomized, controlled trial that included a 12-week exercise intervention of either six sessions (days) per week, two sessions per week, or a sedentary

control group (no exercise). Men and women were randomized separately, with separate allocation sequences generated and maintained by the study statistician. Participants were randomized upon completion of all baseline assessments with no blinding of intervention assignments. 13 of the 19 randomized participants to the six-session group, 17 of the 20 randomized to the two-session group, and 11 of the 14 randomized to the control group were female. Participants were assessed for outcome measures at baseline and after the intervention. Food reinforcement and body composition were assessed 24 to 48 hours after the participant completed their final exercise session of the 12-week intervention while resting energy expenditure was assessed 48 to 72 hours after completing the final exercise session.

**Exercise intervention.** Participants were provided a Polar A-300 heart rate monitor (watch and chest strap, Kempele, Finland) for the duration of the 12-week intervention and instructed to exercise either two or six times per week on their own and were provided access to a fitness center. Participants were instructed to engage in only aerobic exercise either indoors (treadmill, elliptical or cycle ergometer) or outdoors (walking, running, biking) as long as they attained a heart rate at least in heart rate reserve (HRR) zone 1. Participants in the control group were instructed to remain sedentary and return for post-testing 12 weeks later, receiving the exercise intervention after post-testing if they desired. Those in the exercise groups returned to the lab weekly to meet a researcher and download their exercise sessions using the PolarFlow software, which allowed research staff to monitor and track compliance. If a participant was not 85% compliant (completed 85% of expected exercise sessions per month) they were dropped from the study. The downloaded exercise session reports provided the amount of time spent in each heart rate zone, which allowed for the calculation of total energy expended during each exercise session based off individual rates of energy expenditure averaged across each heart rate zone calculated from the graded exercise test with indirect calorimetry performed at baseline and week six. Participants in the two-day per week group were instructed to perform two long exercise sessions per week and encouraged to try to expend 1,000 kcal per session. Participants in the six-day per week group were instructed to keep their sessions to 400 kcal per session. Although most participants in the two-day per week group were not able to attain the 1,000 kcal goal, they still expended significantly greater kcal per session compared to the six-day group. Participants received personalized heart-rate based exercise prescriptions that, if followed, would result in them expending the assigned energy per exercise session. Participants were also provided feedback each week on their energy expenditure of each session of the prior week so they could tailor future exercise sessions. All participants were instructed not to purposely change dietary habits during the intervention, i.e., not begin an energy-restricted diet.

## Assessments

**Liking of foods and hunger (secondary outcome measure).** Participants' liking (hedonic value) of the food options used in the food reinforcement test were assessed using a 10-point scale (1 = "do not like at all" and 10 = "like very much"). Participants sampled three "healthy" snack foods and three "unhealthy" snack foods (Table 1). The most liked food of each category was used during the food reinforcement testing session, completed on a separate day. All participants rated at least a moderate liking (liking score of ≥5) for at least one of the healthy and unhealthy food options. This test was completed once, prior to the baseline food reinforcement test. Hunger and satiety scales were presented prior to each food reinforcement test and used as an additional variable in regression models when predicting changes in food reinforcement. A copy of the VAS scales are provided in the supplementary data.

**Food reinforcement (primary outcome measure).** Reinforcing value of food was assessed by evaluating the amount of operant responding (mouse button presses) a participant

**Table 1. Foods used in the progressive ratio schedules of reinforcement task to measure food reinforcement.**

| | | Kcal/gram | %CHO[1] | %Fat[2] | %Pro[3] | %Sugar[4] |
|---|---|---|---|---|---|---|
| "Unhealthy" food options | Snickers | 4.9 | 50.2 | 43.6 | 6.2 | 41.2 |
| | Nacho Cheese Doritos | 5.40 | 48.0 | 48.0 | 5.2 | 2.7 |
| | Oreo cookies | 4.7 | 60.2 | 38.0 | 4.7 | 35.0 |
| "Healthy" food options | Nutri-grain bar | 3.2 | 73.3 | 20.6 | 6.1 | 80.0 |
| | Dried Banana | 5.2 | 42.8 | 55.7 | 0.02 | 6.8 |
| | Mixed nuts | 6.1 | 12.5 | 75.2 | 12.3 | 0.03 |

[1]%CHO: percent of total kcal derived from carbohydrates

[2]%Fat: percent of total kcal derived from dietary fat

[3]%Protein: percent of total kcal derived from protein

[4]%Sugar: percent of total kcal derived from added sugars.

performed to gain access to each alternative (healthy and unhealthy snack foods, Table 1) [29–31]. The testing environment included two workstations with computers in the same room. One computer had a game that was set-up for participants to earn points towards their highest liked healthy food while the other computer had the same game that participants could use to earn points toward the highest liked unhealthy food. Participants could switch between stations as much as they chose. The computer programs presented a game that mimics a slot machine; a point is earned each time the shapes match. For every five points, a session is completed and the participant received an approximately 60-kcal portion of the reinforcer that was earned (either healthy or unhealthy food). The game was performed until the participant no longer wished to work for access to either food. At first, points were delivered after every four presses, but then the schedule of reinforcement doubled (4, 8, 16, 32, [. . .] 1024) each time five points were earned. For instance, the participant initially had to click the mouse four times to earn each point for schedule one. After the first five points were earned, schedule one was complete and the participant earned a portion of food for the reinforcer earned (healthy or unhealthy food). Then eight clicks were required to earn each of the next five points for schedule two before another portion of the reinforcer was earned. Schedule three required 16 clicks to earn one point, schedule four required 32 clicks to earn one point, and so on [29, 30]. Participants received the food earned after completing the game, which ended when the participant no longer wished to earn points for eating either type of food. This progressive ratio schedule task (Behavioral Choice Task) software was developed at the University of Buffalo, reviewed in [27] and used previously by the current investigative team [31, 37]. Participants were not allowed to take food out of the lab with them, but were not required to consume all of the food they earned in the event a participant earned more food than they could eat in one sitting, although this occurred on only two occasions throughout the study. The food reinforcement tests were conducted two to four hours post-prandial between usual lunch and dinner times when snack foods are likely to be consumed. Similar button pressing tasks have been shown to be valid predictors the reinforcing value of eating behaviors [29, 32–35]. The breakpoint, or $P_{max}$, [30] was the total number of schedules completed for the healthy or unhealthy food. The Relative Reinforcing Value of healthy foods ($RRV_{healthy}$) was the proportion of $P_{max}$ for healthy food compared to the unhealthy food calculated as ($P_{max}$ healthy)/ ($P_{max}$ healthy + $P_{max}$ unhealthy). As constructed, an $RRV_{healthy}$ score over 0.5 indicates a greater reinforcing value for the healthy snack food option relative to unhealthy option, while a $RRV_{unhealthy}$ score of over 0.5 represents a greater reinforcing value of unhealthy food relative to healthy [29, 30]. As a measure of total food reinforcement, $P_{max}$ healthy and $P_{max}$ unhealthy were added together, reflecting the total number of schedules completed for food, or $P_{max}$Total. Thus, $P_{max}$Total

reflects an overall score of total food reinforcement, irrespective of type (healthy or unhealthy). Changes in $P_{max}$ healthy and $P_{max}$ unhealthy were also assessed independently.

**Rate of energy expenditure (secondary outcome measure).** A graded exercise treadmill test was used to determine each participant's rate of energy expenditure at five different heart-rate zones. Oxygen consumed and $CO_2$ produced were analyzed by indirect calorimetry (VMAX Encore Metabolic Cart, Vyaire Medical, Mettawa, IL) which included an integrated 12 lead ECG for monitoring heart rate and used in conjunction with the Trackmaster TMX428 Metabolic cart interfaced treadmill. Upon completion of a five-minute warm-up walking at 0% grade, 3.0 mph, the treadmill grade increased to 2.5% for three minutes. The treadmill grade was then increased every three minutes to produce an approximately 10 beat per minute increase in heart rate from the previous stage with the speed fixed at 3.0 mph. The test continued until a heart rate of 85% HRR was attained or the participant felt they could no longer continue. Energy expenditure (kcal per minute) was determined from the amount of oxygen consumed and $CO_2$ expired using the Weir equation [35]. The average rate of energy expenditure during the last 30 seconds of each stage of the graded exercise test was regressed against the heart rate averaged over the last 30 seconds of the corresponding stage to calculate the rate of energy expenditure at different heart rates. Heart rate zones were calculated using the HRR formula as (220-age)-resting HR * zone % + resting HR [36]. Heart rate Zone 1 ranged from 0% to 25% HRR, Zone 2 corresponded to 26–40% HRR, Zone 3 was 41–58% HRR, Zone 4 was 59–75% HRR, and Zone 5 was 76–90%. Energy expenditure in kcal/min was then averaged across each heart rate zone for determination of energy expenditure per minute for each zone. This test was completed at baseline and week six to recalculate rates of energy expenditure to account for improvements in fitness.

**Resting energy expenditure (secondary outcome measure).** Resting Energy Expenditure (REE) was measured using indirect calorimetry (Quark RMR; Cosmed USA, Chicago, IL) with a ventilated canopy. Before each test, calibrations were performed on the flow meter using a 3.0-L syringe and on the gas analyzers using verified gases of known concentrations. After 30 minutes of quiet rest in the supine position in a dimly lit, temperature-controlled room between 22 and 24 C, REE was measured for 30 minutes. The test was monitored to ensure participants remained awake and between 0.8 and 1.2% feCO2. Criteria for a valid REE was a minimum of 15 minutes of steady state, determined as a <10% fluctuation in oxygen consumption and <5% fluctuation in respiratory quotient. The Weir equation (36) was used to determine REE from the measured oxygen consumption and $CO_2$ production. Participants completed the baseline REE assessment prior to the exercise test and 36–72 hours after their final exercise session of the intervention.

**Body composition (primary outcome measure).** Body composition was measured using a GE Lunar iDXA machine prior to the exercise test. The iDXA technique allows the non-invasive assessment of soft tissue composition by region with a precision of 1–3% [37]. A total body scan was conducted with participants lying supine on the table and arms positioned to the side. Most scans were completed using the thick mode suggested by the software as participants were overweight to obese. All scans were analyzed using GE Lunar enCORE Software (13.60.033). Automatic edge detection was used for scan analyses. The machine was calibrated before each scanning session, using the GE Lunar calibration phantom. Outcome measures included kg of body fat mass (FM), kg of fat-free mass (FFM)- which included body water, bone mineral content, and dry lean mass, and percent FM.

## Analytic plan

Differences in changes in FM, FFM, REE, $RRV_{healthy}$, $RRV_{unhealthy}$, $P_{max}$ healthy, $P_{max}$ unhealthy and total food reinforcement ($P_{max}Total$) between groups were determined via one-

way ANOVA. Change scores were also tested if significantly different from zero via T-tests for each group, separately. Shapiro-Wilk tests and inspection of histograms revealed a non-normal distribution for changes in $P_{max}$Total, thus differences in $P_{max}$Total between groups were also assessed via Kruskal Wallis Test. Spearmen correlation analysis was performed between changes in $P_{max}$Total, hunger, FM, FFM, REE, and exercise variables including exercise frequency (group) and exercise energy expenditure per week. Significant correlations were further assessed with quantile regression, using change in $P_{max}$Total as the dependent variable. All analyses were performed in IBM SPSS Version 26 (IBM corporation, Armonk, New York).

**Power analysis.** The present analysis is a secondary outcome of a study originally designed to test effects of exercise dose on energy compensation, powered to detect significant differences in body fat loss between groups to draw conclusions regarding energy compensation in a clinically relevant scenario, that is, a scenario in which individuals who are overweight to obese lose a significant amount of body weight. We previously demonstrated significant differences (1.7 kg) in body fat loss in groups exercising to expend 3,000 kcal per week vs. 1,500 kcal per week for 12 weeks [37]. Using an 80% power and 95% confidence level, 13 participants per group were needed to detect a significant change in body fat loss from baseline to post intervention with a standard deviation of 2.3.

## Results

Baseline measures of food measures, BMI, body composition, REE, and demographics (% female) are presented in Table 2. Change scores (post value minus baseline value) for outcome variables and exercise energy expenditure are presented in Table 3. The six-day per week group expended more energy than the two-day per week group, which contributed to only the six-day group significantly decreasing percent body weight and FM. Change in FM was greater in both intervention groups compared to control with no differences in FFM change. There were no changes in $P_{max}$Total, $P_{max}$ healthy, $P_{max}$ unhealthy, REE, or RRV of healthy or

**Table 2. Demographics, body composition, and food reinforcement measures for all randomized participants at baseline.**

| | Six-day per week group N = 19 | Two-day per week group N = 20 | Control N = 14 |
|---|---|---|---|
| Sex (% female) | 68.4 | 85.0 | 78.8 |
| BMI[1] | 29.0 ± 2.87 | 30.51 ± 3.47 | 29.36 ± 2.87 |
| FM[2] | 31.25 ± 8.58 | 35.58 ± 6.55 | 30.27 ± 6.42 |
| FFM[3] | 48.50 ± 9.68 | 48.91 ± 9.35 | 43.77 ± 5.92 |
| Body Fat % | 37.96 ± 6.87 | 41.29 ± 4.41 | 39.68 ± 4.11 |
| REE[4] | 1505.8 ± 48.13 | 1666.9 ± 76.55 | 1546.40 ± 50.59 |
| RRV$_{Healthy}$[5] | 0.54 ± 0.32 | 0.62 ± 0.29 | 0.62 ± 0.31 |
| RRV$_{Unhealthy}$[6] | 0.46 ± 0.32 | 0.38 ± 0.30 | 0.38 ± 0.31 |
| $P_{max}$Total[7] | 65.00 ± 73.77 | 78.25 ± 97.59 | 102.00 ± 108.97 |

Data are mean ± SD

[1]Body Mass Index, kg/m$^2$

[2]FM: Fat Mass, kg

[3]FFM: Fat Free Mass, kg

[4]REE: Resting energy expenditure (Kcal/24 hours)

[5]RRV$_{Healthy}$: Relative Reinforcing value of healthy food, assessed via progressive ratio schedule of reinforcement task and calculated as last schedule completed for healthy food / total schedules completed

[6]RRV$_{Unhealthy}$: Relative Reinforcing value of unhealthy food, assessed via progressive ratio schedule of reinforcement task and calculated as the last schedule completed for unhealthy food / total schedules completed

[7]$P_{max}$Total: Total schedules completed for food, both healthy and unhealthy, in the progressive ratio schedule of reinforcement task

**Table 3. Resulting data from the 12-week exercise intervention between groups that exercised either six or two days per week and a sedentary control group.**

| | Six-day per week group N = 15 | Two-day per week group N = 17 | Control N = 11 |
|---|---|---|---|
| ExEE/week[1] | 2,753.5 ± 561.2* | 1,490.7 ± 503.4* | 0* |
| Percent weight loss[2] | -1.48 ± 2.48^ | -0.84 ± 2.72 | +1.45 ± 3.71 |
| $\Delta$FM[3] | -1.82 ± 1.51^# | -0.64 ± 0.95* | 0.98 ± 2.62^* |
| $\Delta$FFM[4] | 0.38 ± 1.39 | -0.04 ± 0.23 | -0.06 ± 1.36 |
| $\Delta$REE[5] | 39.18 ± 151.5 | -38.0 ± 246.5 | -1.13 ± 259.0 |
| $\Delta$RRV healthy food[6] | 0.12 ± 0.35 | -0.05 ± 0.41 | -0.01 ± 0.20 |
| $\Delta$RRV junk food[7] | -0.12 ± 0.35 | 0.11 ± 1.86 | 0.13 ± 0.36 |
| $\Delta P_{max}$Total[8] | 20.36 ± 141.7 | -39.38 ± 93.31 | -14.55 ± 78.87 |
| $\Delta P_{max}$healthy[9] | 33.21 ± 131.07 | -36.47 ± 78.78 | -33.25 ± 56.83 |
| $\Delta P_{max}$unhealthy[10] | -12.86 ± 54.94 | -0.88 ± 23.84 | 21.36 ± 51.17 |

Data are mean ± SD

*, ^: like letters indicate significant differences between groups, $P \leq 0.05$.

#Significant change over time (change different from zero) $P \leq 0.05$.

[1]ExEE/week: Exercise energy expenditure (in kilocalories) per week.

[2]Percent weight loss: kg of weight change (12-week value minus baseline value) / baseline body weight in kg

[3]$\Delta$FM: kg of fat mass change: (12-week kg of body fat mass minus baseline kg body fat)

[4]$\Delta$FFM: kg of fat-free mass change (12-week kg of fat-free mass minus baseline kg of fat free mass)

[5]$\Delta$REE: Changes in REE (12-week value minus baseline value) in kcal/24 hrs.

[6]$\Delta$RRV$_{healthy}$: changes in relative reinforcing value of healthy food assessed via progressive ratio schedule of reinforcement task, calculated as 12-week RRV$_{healthy}$−baseline RRV$_{healthy}$

[7]$\Delta$RRV$_{unhealthy}$: changes in relative reinforcing value of unhealthy food assessed via progressive ratio schedule of reinforcement task, calculated as 12-week RRV$_{unhealthy}$−baseline RRV$_{unhealthy}$

[8]$\Delta P_{max}$Total: changes in total schedules completed for food (healthy + unhealthy) during the progressive ratio schedule of reinforcement task.

[9]$\Delta P_{max}$healthy: changes in total schedules completed for healthy food option during the progressive ratio schedule of reinforcement task.

[10]$\Delta P_{max}$unhealthy: changes in total schedules completed for unhealthy food option during the progressive ratio schedule of reinforcement task.

unhealthy food over time or between groups (Table 3). All variables were of normal distribution with the exception of changes in $P_{max}$Total (Shapiro-Wilk P<0.05, Skewness: 1.24, SE = 0.37; Kurtosis: 9.26, SE = 0.72; histogram included in supplementary data). Thus, a Kruskal-Wallis Test was used to confirm no difference between groups for changes in $P_{max}$Total (Table 4). Spearmen correlation analysis indicated that $P_{max}$Total negatively correlated with changes in FFM (P = 0.03), with no correlations between changes in FM, REE, hunger or exercise parameters (weekly energy expenditure, exercise group). Positive correlations (both when assessed via Pearson or Spearmen) were observed between FM change, exercise energy expenditure per week, and exercise group (all P<0.1). Pearson correlations demonstrated a positive correlation between changes in REE and changes in hunger (P = 0.02). Quantile regression analysis predicting changes in $P_{max}$Total is presented in Table 5, demonstrating changes in FFM to be the only significant predictor for changes in $P_{max}$Total when controlling for weekly

**Table 4. Kruskal Wallis Test for changes total food reinforcement ($\Delta P_{max}$Total, non-normally distributed) between groups that exercised either six or two days per week and a sedentary control group.** Data are mean rank.

| | Six-day per week group N = 15 | Two-day per week group N = 17 | Control N = 11 | $\chi^2(2)$ |
|---|---|---|---|---|
| $\Delta P_{max}$Total[1] | 24.46 | 18.75 | 1.86 | 1.872 |

P = 0.392

[1]$\Delta P_{max}$Total: changes in total schedules completed for food (healthy + unhealthy) assessed during the progressive ratio schedule of reinforcement task

**Table 5. Quantile regression models predicting changes in total food reinforcement ($P_{max}$Total) among participants who exercised for 12 weeks at a frequency of either two or six days per week.**

| Effect | β | SE | P |
|---|---|---|---|
| **Full model of all predictors** | | | |
| Intercept | -161.7 | 151.3 | 0.32 |
| ExEE/week[1] | -0.05 | 0.06 | 0.43 |
| Exercise Frequency[2] | 68.8 | 75.4 | 0.39 |
| ΔFM[3] | -26.6 | 21.4 | 0.25 |
| ΔFFM[4] | -72.6 | 22.7 | 0.01 |
| ΔREE[5] | -0.04 | 0.18 | 0.84 |
| ΔHunger[6] | 1.69 | 1.83 | 0.38 |
| **Reduced model** | | | |
| Intercept | -36.2 | 21.7 | 0.11 |
| ΔFM[3] | -7.01 | 11.2 | 0.54 |
| ΔFFM[4] | -46.2 | 18.5 | 0.02 |
| ΔREE[5] | 0.06 | 0.11 | 0.54 |

[1]ExEE/week: Exercise energy expenditure (in kilocalories) per week

[2]Exercise Frequency: Participants were randomly assigned to exercise 6 days per week or 2 days per week.

[3]ΔFM: kg of fat mass change: (12-week kg of body fat mass minus baseline kg body fat)

[4]ΔFFM: kg of fat-free mass change (12-week kg of fat-free mass minus baseline kg of fat free mass)

[5]ΔREE: Changes in REE (12-week value minus baseline value) in kcal/24 hrs.

[6]ΔHunger: Hunger assessed prior to each progressive ratio schedules of reinforcement task (1–10 scale).

exercise energy expenditure, exercise frequency per week (group), and changes in FM, REE and hunger. A reduced model is also presented, removing independent variables that were correlated with each other to account for collinearity, thus leaving changes in FFM, FM and REE as the independent variables with only changes in FFM remaining a significant predictor. Mediation analysis did not reveal significant mediation effects for changes in REE on changes in FFM when predicting changes in $P_{max}$Total (Sobel test statistic: <0.01, SE = <0.01, P = 0.99).

## Discussion

Most people have an enormous capacity to increase their energy expenditure to promote a negative energy balance. Depending on an individual's aerobic fitness, exercise intensities can be maintained for prolonged periods at two- to 16-fold above resting rates of energy expenditure. As such, 250 to 2500 kcal can be expended during a single exercise session resulting in an acute energy deficit that can be repeated across days, prompting many individuals to turn to exercise for obesity treatment [38, 39]. This has led the American College of Sports Medicine to issue separate recommendations to either maintain health [40] or support weight loss through exercise (5). For these reasons, exercise has become the most common weight loss approach amongst those attempting to lose weight, with a 65% prevalence rate [41]. As the obesity epidemic continues to escalate in America and other developed countries, the number of individuals seeking to lose weight is also expected to increase, thereby increasing the number of individuals using exercise as their weight loss method of choice. This is despite the lack of consistency in weight loss outcomes from exercise interventions and lack of long-term weight loss maintenance [42]. For this reason, additional research is needed to determine factors influencing weight loss from an exercise program and methods to improve exercise's utility as a weight loss treatment.

The current study is an initial investigation on how exercise may influence eating behaviors through a mechanism separate from hunger (reward-driven feeding), an alternative approach that has received much less attention, although still justifiable, as research on exercise's effects on hunger are inconsistent with many concluding little to no effects [15–18]. Reward-driven feeding may therefore have a greater influence on the increased drive to eat while in an energy-deficit induced by exercise. However, very little research has focused on the effects exercise may have on food reinforcement and how this may alter eating behaviors, which is a plausible hypothesis in that food and exercise reinforcement are both driven by the meso-accumbal dopamine system. The central dopamine system is essential for experiencing reward and assimilating information about energy balance (i.e. the dopamine hypothesis of reward), originally proposed to explain drug addiction [23, 24, 43]. Indeed, genetic polymorphisms that control dopamine uptake and transport have been linked to both exercise and eating reinforcement [32, 44–46], possibly explaining the cross-talk observed between eating and exercise reward in rats [25]. Research on this topic in humans, however, is sparse, with one study demonstrating no changes in food reward after an acute bout of exercise [47] and another demonstrating decreases in the reinforcing value of high-energy dense foods with increases in the reinforcing value of low-energy dense foods after two weeks of aerobic exercise [48]. These findings may indicate longer duration exercise interventions are needed to increase food reinforcement and thus influence eating behavior. This would be in line with previous research demonstrating changes in exercise reinforcement occur only after large-dose exercise interventions [49, 50]. It is also plausible that a large energy deficit needs to be created (larger than what can be created in one exercise bout or a two-week intervention) to increase reward-driven feeding. In this light, a yet to be determined metabolic signal that feeds back on the central dopamine system to instill greater food reinforcement when energy stores are low may exist. The notion that the reinforcing value of low energy dense foods increased after short-term exercise owe to the possibility that individuals may be more motivated to eat only certain (healthier) foods when partaking in exercise, which is in line with research pointing to one health behavior change having a spillover effect on other behaviors [48, 51]. For this reason, the present study analyzed the reinforcing value of two different types of food independently and constructed a measure of total food reinforcement (total number of schedules completed for all food). In this scenario, one may decrease their reinforcing value of unhealthy foods and increase their reinforcing value of healthy foods to result in a gross increase in food reinforcement if the increase in healthy foods was greater than the decrease in unhealthy. This appeared to have occurred in the present study among the six-day per week group, although the increases in this group did not reach statistical significance due to large variability (minimum and maximum values for changes in $P_{max}$Total ranged from -340 to 440). Being a secondary aim of the original trial (assessing mechanisms of energy compensation using different doses of exercise) this analysis may not have been adequately powered to detect changes in total responses for food and thus would be the logical next step for future interventions. This is also evident when focusing on the changes in $P_{max}$ healthy and $P_{max}$ unhealthy, where the six-day per week group completed over 12 schedules less for unhealthy foods and 33 schedules more for healthy food post-intervention, although none of these changes were deemed significant.

Despite the lack of significant changes in $P_{max}$Total after the exercise intervention, regression analysis revealed that changes in FFM were a significant independent predictor for changes in $P_{max}$Total, whereas greater losses in FFM produced more responding for food. This held when controlling for exercise energy expenditure (kcal per week), exercise frequency (sessions per week), and changes in FM, REE and hunger. In this full model, several variables were correlated with each other, presenting the likelihood of collinearity causing an inflation of regression coefficients. Specifically, changes in FM, weekly exercise energy expenditure, and

exercise group were all correlated. Thus, we removed both weekly exercise energy expenditure and exercise group while keeping changes in FM in the reduced model as FM change was likely the result of exercising at greater frequencies and expending more energy per week. Interestingly, changes in REE were positively correlated with changes in hunger, whereas those who had greater increases in hunger had greater increases in REE. Such correlations have been demonstrated before, with the notion that FFM determines REE to influence hunger and food intake [52]. Although others have concluded REE, and not FFM, influences hunger and food intake [53, 54], the present analysis indicate that FFM is the primary driver of food reinforcement, thus we included changes in REE in the final model due to the known link between FFM, REE, and food intake regulation [55]. Therefore, the reduced model included changes in FM, FFM, and REE as predictors of changes in total food reinforcement ($P_{max}$Total), where changes in FFM remained the only significant predictor, indicating changes in FFM after a 12-week exercise for weight loss intervention influences food reinforcement to the greatest degree. This is supported by recent findings, indicating FFM is associated with several brain regions involved in the homeostatic control of food intake, suggesting a centrally mediated mechanism whereby FFM influences eating behaviors [56]. This has also been observed in the classic Minnesota Starvation Experiments, where the FFM deficit, independently of FM deficit, predicted the degree of hyperphagia that occurred during post-starvation *ad libitum* refeeding, which continued until FFM was repleted, often well after FM was restored [57, 58].

The present findings, demonstrating FFM deficits were the root cause in the increase in operant responding for food after 12-weeks of exercise, extend previous research by suggesting a direct mechanism that may be prompting increases in energy intake when FFM is depleted. This is an important consideration for present weight-loss recommendations, as many individuals are utilizing aerobic exercise to induce the energy deficit required for weight loss. This aerobic exercise-induced weight loss, especially when coupled with energy restriction, almost always produces concurrent losses in FFM [55, 59]. Although decreases in FFM, if accompanied by a larger decrease in FM, is often overlooked as many markers of cardiometabolic health are often improved. However, as indicated by the present study and supported by others, declines in FFM may have the unintended consequence of increasing food reinforcement, causing overeating to promote the return to baseline FFM. Unfortunately, when returning to baseline FFM levels, "fat overshooting" often occurs, where FM is increased beyond baseline levels [55, 60]. Therefore, one can argue that exercise-for weight loss interventions should include resistance exercise to promote, or at least maintain, FFM when attempting weight loss [61]. Maintaining FFM, in this light, would theoretically attenuate the increase in food reinforcement and improve long-term weight-loss success with exercise. Future interventions assessing the effects of resistance exercise and food reinforcement following weight loss may add to current exercise recommendations to increase the effectiveness of exercise as a weight loss strategy. It may also be interesting to investigate how the reinforcing value of food changes during an exercise intervention to determine how may exercise sessions, how large of an energy deficit, or how much FFM needs to be lost to render food more rewarding. It may also be interesting to investigate how long it may take for food reinforcement to return to baseline levels, or if it remains elevated long after ceasing exercise and returning to energy balance.

This study is not without limitations. Liking of the test foods was not assessed as baseline liking scores did not influence food reinforcement, but if may have been interesting to see how or if liking of these tests foods changed as a result of the exercise intervention. Ab libitum energy intake would be best assessed in an inpatient feeding design where all of participants' meals and snacks are consumed in a controlled environment and recorded by research staff to prevent the known under-reporting that often occurs with self-reported dietary intake [62]. Without this, the present study does not have the exact amount of energy consumed, how

dietary intake changed during the exercise program, and how these changes aligned with the changes in food reinforcement. Due to the known influence restrictive diets play on food reinforcement [37], participants were inscructed not to purposely change their habitual diets (i.e. start a weight loss diet). Due to the modest decrease in weight loss, we do not believe participants purposely restricted their energy intake. Contrary to that, it is likely that participants consumed more food to compensate for the energy expended, although we do not know this for sure without rigorous dietary intake assessment methods. As previously noted, the analysis, being a secondary aim of a larger study, was likely underpowered to detect significant differences in change scores (post value minus baseline value) in food reinforcement and likely affected our mediation analysis, an issue future studies must address. The duration of the intervention (12 weeks) is similar to most exercise interventions in this area of research, although a longer period may have been required to detect significant changes in food reinforcement and changes in percent weight loss among the two-day per week group. We also did not monitor adherence of the control group, although we believe each remained sedentary as they did not significantly change body weight after 12 weeks. Although without accelerometry data, we cannot be sure those in the control group did not begin exercising on their own. Participants were also only provided their most liked food out of five choices for each the healthy and unhealthy options for the RRV assessment. If participants were provided with their favorite food choice they may have responded differently. Additionally, out of the 44 participants who completed the current study, 40 were white, thus limiting the generalizability to other race/ethnic groups. This study also was not designed to detect sex differences and included an unbalanced sample of females; thus, sex effects cannot be drawn.

## Conclusions

The present analysis offers an initial look into an alternative explanation as to why exercise programs are often only marginally effective for weight loss. Our findings indicate that there is great variability in individuals' change in food reinforcement after a 12-week aerobic exercise intervention, and those who did increase their food reinforcement were also those who lose the greatest amount of FFM post-intervention, even when controlling for other variables such as the energy expended during the intervention or the frequency of exercise (sessions per week). From these findings we can draw two primary conclusions. First, it appears a large sample size is required to better elucidate food reinforcement changes after exercise, potentially taking sex effects into account. Second, it appears preventing the loss in FFM would be a valuable piece to a weight loss program, hinting at the potential for resistance training or increasing dietary protein intake as important adjunct therapy that may be the topic of future research.

## Supporting information

**S1 Checklist.**
(PDF)

**S1 File. PARQ, physical activity readiness questionnaire.** Provided to participants at baseline to ensure they were healthy enough to engage in exercise without physician consent.
(PDF)

**S2 File. Healthy history questionnaire.** Provided to participants at baseline to ensure the met certain inclusion criteria regarding current or recent health issues.
(PDF)

**S3 File. VAS scales, visual analog scales.** Scale used to assess hunger and desire to eat prior to each RRV test.
(PDF)

**S4 File. Liking of food.** Questionnaire used to assess liking of study foods to determine which will be used in the RRV test.
(PDF)

**S5 File. Study protocol.** IRB-approved protocol for present trial.
(PDF)

**S6 File. Simple histogram for delta P$_{max}$Total.** Histogram to depict changes in P$_{max}$Total.
(PDF)

# Acknowledgments

The authors would like to thank the staff of the University of Kentucky's Center for Clinical and Translational Science (CCTS), who aided in recruitment and assessments throughout the study. The authors would also like to thank the participants for their time and effort in performing the intervention and willingness to be assessed, and the department of Dietetics and Human Nutrition for funding the present study.

# Author Contributions

**Conceptualization:** Kyle D. Flack.

**Data curation:** Kyle D. Flack, Harry M. Hays, Jack Moreland.

**Formal analysis:** Kyle D. Flack.

**Funding acquisition:** Kyle D. Flack.

**Investigation:** Kyle D. Flack, Harry M. Hays, Jack Moreland.

**Methodology:** Kyle D. Flack, Jack Moreland.

**Project administration:** Kyle D. Flack, Harry M. Hays, Jack Moreland.

**Resources:** Kyle D. Flack.

**Software:** Kyle D. Flack.

**Supervision:** Kyle D. Flack.

**Validation:** Kyle D. Flack.

**Visualization:** Kyle D. Flack.

**Writing – original draft:** Kyle D. Flack.

**Writing – review & editing:** Kyle D. Flack.

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
