## [Decision Letter · Decision Letter 0]

1 Apr 2020

PONE-D-20-01440

The Consequences of Exercise-Induced Weight Loss on Food Reinforcement, A Randomized, Controlled Trial

PLOS ONE

Dear Dr. Flack,

Thank you for submitting your manuscript to PLOS ONE. After careful consideration, we feel that it has merit but does not fully meet PLOS ONE’s publication criteria as it currently stands. Therefore, we invite you to submit a revised version of the manuscript that addresses the points raised during the review process.

The Reviewers have several minor comments especially pertinent to Methods and Discussion of the manuscript that need detailed responses and clarifications.

We would appreciate receiving your revised manuscript by May 16 2020 11:59PM. To enhance the reproducibility of your results, we recommend that if applicable you deposit your laboratory protocols in protocols.io, where a protocol can be assigned its own identifier (DOI) such that it can be cited independently in the future. For instructions see: http://journals.plos.org/plosone/s/submission-guidelines#loc-laboratory-protocols

We look forward to receiving your revised manuscript.

Kind regards,

Maciej S. Buchowski

Academic Editor

PLOS ONE

Journal Requirements:

1. Please explain any discrepancy between the trial protocol and the manuscript; for example, we note that, while the manuscript states that participants aged 18- 49  were included, in the registry and in the protocol the age limit is set at 40 years old .

2. Please include additional information regarding the survey or questionnaire used in the study and ensure that you have provided sufficient details that others could replicate the analyses. For instance, if you developed a questionnaire as part of this study and it is not under a copyright more restrictive than CC-BY, please include a copy, in both the original language and English, as Supporting Information. Moreover, please include more details on how the questionnaire was pre-tested, and whether it was validated.

3. Please note that according to our submission guidelines (http://journals.plos.org/plosone/s/submission-guidelines), outmoded terms and potentially stigmatizing labels should be changed to more current, acceptable terminology. For example: “Caucasian” should be changed to “white” or “of [Western] European descent” (as appropriate).

4. Our internal editors have looked over your manuscript and determined that it is within the scope of our Determinants, Consequences and Management of Obesity  Call for Papers. This collection of papers is headed by a team of Guest Editors for PLOS ONE:Rachel Nugent and Pratibha V. Nerurkar. Additional information can be found on our announcement page: https://collections.plos.org/s/obesity-one.  

If you would like your manuscript to be considered for this collection, please let us know in your cover letter and we will ensure that your paper is treated as if you were responding to this call. If you would prefer to remove your manuscript from collection consideration, please specify this in the cover letter.

5.  Thank you for including the following funding information within your acknowledgements section; "The project described was supported by the NIH National Center for Advancing 410 Translational Sciences through grant number UL1TR001998"

6.  We note that you have indicated that data from this study are available upon request. PLOS only allows data to be available upon request if there are legal or ethical restrictions on sharing data publicly. For information on unacceptable data access restrictions, please see http://journals.plos.org/plosone/s/data-availability#loc-unacceptable-data-access-restrictions.

7. Please upload a copy of Figure 1, to which you refer in your text on page 13. If the figure is no longer to be included as part of the submission please remove all reference to it within the text.

Reviewers' comments:

Reviewer's Responses to Questions

**Comments to the Author**

1. Is the manuscript technically sound, and do the data support the conclusions?

Reviewer #1: Yes

Reviewer #2: Yes

2. Has the statistical analysis been performed appropriately and rigorously? 

Reviewer #1: Yes

Reviewer #2: Yes

3. Have the authors made all data underlying the findings in their manuscript fully available?

Reviewer #1: Yes

Reviewer #2: Yes

4. Is the manuscript presented in an intelligible fashion and written in standard English?

Reviewer #1: Yes

Reviewer #2: Yes

5. Review Comments to the Author

Reviewer #1: Overall

This paper builds upon both theoretical considerations, the impact of physical activity upon dietary reinforcement, while also offering practical advice for those interested in losing weight via physical activity, maintain lean mass via resistance training. The overall paper is well written and adds to the literature. The additions I would suggest are as follows:

Introduction:

Line 40: add that exercise also essential to maintaining healthy weight after loss. Considerable research evidence supports this.

Line 62: “seek rewards FOR exercising …”

Methods:

Line 84: Can you say “increase food reinforcement” if measuring 2 food types?

Line 146: Did any participants report an atypical diet (keto, Atkins etc.) diet or that they were currently dieting to lose weight at baseline?

Line: 191 spell-check of “unhealthy”

Line 186-191: Pmax can be calculated for healthy and unhealthy separately. Specify that Pmax_total is for both added together. Also, why did you only examine Pmax total and not the separate Pmaxs for each food group?

Mention that participants’ favorite food (healthy or unhealthy) may not have been offered and they would respond differently to different options

Did you examine proportion of food earned during the RRV task to food consumed afterwards? Those results may be interesting.

Results

Line 237-238: Because changes in body composition are so important for study results, please include body fat percentage, and kg of fat mass and lean mass in Tables 2.

Line 264: Unhealthy food, to keep terminology consistent

Discussion:

RRV task and DXA took place very shortly after end of a very intense training regime. How might results change if participants were given opportunity to taper and/or rest before post-intervention results?

Any indication whether or not participants change their diets, either intentionally or unintentionally?

Reviewer #2: This is a well-crafted study that examines a novel and important topic. I enjoyed reading it. I have listed some suggestions, by section, below. While I believe you can handle these issues relatively easily, I do think they will strengthen the manuscript and should be addressed. If these issues are addressed, I believe the manuscript warrants publication. Well done!

Introduction

“The most prevalent mechanism responsible for maintaining energy homeostasis during an exercise program is increases in energy intake, largely due to the fact that the rate of energy intake far exceeds the rate of energy expenditure” while I agree with this point, it may be worthwhile to discuss metabolic changes that can occur with weight loss. At least mention adaptive thermogenesis as a possible mechanism complicating weight loss from exercise. It’s possible that alterations in food reinforcement may be part of the adaptive thermogenesis response.

Methods

“… 52 completed all baseline tests and were randomized into one of three groups…” I think it’s a good idea to list the three groups parenthetically here. Revie to “… 52 completed all baseline tests and were randomized into one of three groups (six exercise sessions per week, two weekly sessions, sedentary control)…”

“The study was a randomized, controlled trial that included a 12-week exercise intervention of either six sessions (days) per week, two sessions per week, or a sedentary control group (no exercise) blocked on gender” list the number of males and females in each group.

How was compliance of the sedentary group assessed?

What were some of the exercise options available top your exercise groups? Did they only do cardio (e.g., treadmill, bike) or were they able to do resistance training?

Why was liking only assessed at baseline? Changes in hedonics as a result of the intervention would have been interesting to look at as well as RRV.

Interesting uses of Pmax for the RRV task. I like how you quantified RRV but, out of curiosity, why not just use raw Pmax scores for healthy versus unhealthy foods as your measure of RRV?

“Change scores were also tested if significantly different from zero via T-tests.” Were these done for each group separately?

“Differences in changes in…” revise to “Differences in change scores (post value minus baseline value) in…”

Discussion

“The present findings, demonstrating FFM deficits are the root cause in the increase…” this language is too definitive for my liking. Revise to “The present findings, demonstrating FFM deficits were the root cause in the increase…” or something similar.

“Our findings indicate that there is great variability in individuals’ change in food reinforcement after a 12-week aerobic exercise intervention, and those who do increase their food reinforcement are those who lose the greatest amount of FFM post-intervention…” Again this reads as too definitive. I think it’s important to describe the results only within the context of the present study. Stating “those who do increase their food reinforcement are those who lose the greatest amount of FFM post-intervention” sounds like you are concluding this is what happens with weight loss for everyone. While that may be true, you cannot conclude that from this single study. I know this seems like a small detail, but it is important to remain tentative. Revise to “Our findings indicate that there was great variability in individuals’ change in food reinforcement after a 12-week aerobic exercise intervention, and those who did increase their food reinforcement were also those who lost the greatest amount of FFM post-intervention…” This way you are only talking about your results.

6. PLOS authors have the option to publish the peer review history of their article (what does this mean?). If published, this will include your full peer review and any attached files.

Reviewer #1: No

Reviewer #2: Yes: Jacob E Barkley

---

## [Author Response · Author response to Decision Letter 0]

14 Apr 2020

Response to Reviewers

We would like to thank the reviewers and Dr. Buchowski for their time and effort into providing their important feedback. We have carefully addressed each comment and feel that this manuscript has improved significantly. Specific comments are addressed below:

1. Please explain any discrepancy between the trial protocol and the manuscript; for example, we note that, while the manuscript states that participants aged 18- 49 were included, in the registry and in the protocol the age limit is set at 40 years old. 

 This was incorrectly stated in the manuscript; we have now corrected this on line 91.

2. Please include additional information regarding the survey or questionnaire used in the study and ensure that you have provided sufficient details that others could replicate the analyses. For instance, if you developed a questionnaire as part of this study and it is not under a copyright more restrictive than CC-BY, please include a copy, in both the original language and English, as Supporting Information. Moreover, please include more details on how the questionnaire was pre-tested, and whether it was validated.

 There were no survey/questionnaires used to gather data on primary or secondary outcomes. Participants completed questionnaires at baseline to demonstrate they were healthy enough to participate in the exercise intervention. Participants also rated their liking of the study foods at baseline and rated their hunger and desire to eat on visual analog scales prior to each RRV test. A copy of each is included as supporting information as requested. 

3. Please note that according to our submission guidelines (http://journals.plos.org/plosone/s/submission-guidelines), outmoded terms and potentially stigmatizing labels should be changed to more current, acceptable terminology. For example: “Caucasian” should be changed to “white” or “of [Western] European descent” (as appropriate).

 We changed “Caucasian” to “white” on line 487 and removed other potentially stigmatizing labels in this section.

4. Our internal editors have looked over your manuscript and determined that it is within the scope of our Determinants, Consequences and Management of Obesity Call for Papers. This collection of papers is headed by a team of Guest Editors for PLOS ONE:Rachel Nugent and Pratibha V. Nerurkar. Additional information can be found on our announcement page: https://collections.plos.org/s/obesity-one. 

If you would like your manuscript to be considered for this collection, please let us know in your cover letter and we will ensure that your paper is treated as if you were responding to this call. If you would prefer to remove your manuscript from collection consideration, please specify this in the cover letter.

 We have specified in the cover letter that we would like to include this manuscript in the Determinants, Consequences and Management of Obesity call.

5. Thank you for including the following funding information within your acknowledgements section; "The project described was supported by the NIH National Center for Advancing 410 Translational Sciences through grant number UL1TR001998"

 We have removed the funding-related text from the manuscript. We would like our funding statement to read: “The project described was supported by the NIH National Center for Advancing Translational Sciences through grant number UL1TR001998. The content is solely the responsibility of the authors and does not necessarily represent the official views of the NIH. The funders had no role in study design, data collection and analysis, decision to publish, or preparation of the manuscript.” 

6. We note that you have indicated that data from this study are available upon request. PLOS only allows data to be available upon request if there are legal or ethical restrictions on sharing data publicly. For information on unacceptable data access restrictions, please see http://journals.plos.org/plosone/s/data-availability#loc-unacceptable-data-access-restrictions.

 There are no legal or ethical restrictions on sharing our data, thus we have deposited our data in a public repository and provided the DOI on Line 511 of the manuscript.

7.Please upload a copy of Figure 1, to which you refer in your text on page 13. If the figure is no longer to be included as part of the submission please remove all reference to it within the text.

 Our figure has been uploaded; we apologize for overlooking this in the initial submission.

 We have included captions for our supporting information as requested at the end of our manuscript, lines 674-682

Reviewer #1: Overall

This paper builds upon both theoretical considerations, the impact of physical activity upon dietary reinforcement, while also offering practical advice for those interested in losing weight via physical activity, maintain lean mass via resistance training. The overall paper is well written and adds to the literature. The additions I would suggest are as follows:

Introduction:

Line 40: add that exercise also essential to maintaining healthy weight after loss. Considerable research evidence supports this.

 We certainly agree with this statement and included “and weight loss maintenance” in this statement as our reference (#5) also backs this (line 42).

Line 62: “seek rewards FOR exercising …”

 This has been edited as suggested, now line 63.

Methods:

Line 84: Can you say “increase food reinforcement” if measuring 2 food types?

 This is a fine point and we further clarified that we hypothesized food reinforcement for specifically high energy dense foods available would increase (line 85-86).

Line 146: Did any participants report an atypical diet (keto, Atkins etc.) diet or that they were currently dieting to lose weight at baseline?

 Participants were not dieting to lose weight at baseline. This was an inclusion criterion, along with being weight stable. We realize this was not included in the participant characteristics section and now include both of these criteria (line 101-102)

Line: 191 spell-check of “unhealthy” 

 This has been corrected, now line 210.

Line 186-191: Pmax can be calculated for healthy and unhealthy separately. Specify that Pmax_total is for both added together. Also, why did you only examine Pmax total and not the separate Pmaxs for each food group?

The last two sentences of this section state “Pmax healthy and Pmax unhealthy were added together, reflecting the total number of schedules completed for food, or PmaxTotal. Thus, PmaxTotal reflects an overall score of total food reinforcement, irrespective of type (healthy or unhealthy)” There were no differences in changes in Pmax healthy or Pmax unhealthy, we specified this in Table 3, results (line 284), and touched on it in the discussion (lines 409-412). 

Mention that participants’ favorite food (healthy or unhealthy) may not have been offered and they would respond differently to different options

 We have included this point when addressing limitations of the present study, lines 484-486..

Did you examine proportion of food earned during the RRV task to food consumed afterwards? Those results may be interesting.

 There were only two occasions where participants did not consume all the food they earned in the RRV task, thus these analyses were not preformed. We have stated this more specifically on line 202. 

Results

Line 237-238: Because changes in body composition are so important for study results, please include body fat percentage, and kg of fat mass and lean mass in Tables 2.

 We have now included baseline values for body fat %, FM and FFM in table 2.

Line 264: Unhealthy food, to keep terminology consistent

 This change has been made, now line 285.

Discussion:

RRV task and DXA took place very shortly after end of a very intense training regime. How might results change if participants were given opportunity to taper and/or rest before post-intervention results? 

 This is an interesting question and we have expanded on this regarding when RRV testing is done in relation to the exercise intervention, lines 456-461

Any indication whether or not participants change their diets, either intentionally or unintentionally? 

We have included, lines 470-475, a couple sentences about how we believe they did not intentionally engage in restrictive eating behaviors but likely increased their food intake in response to the exercise intervention.

Reviewer #2: This is a well-crafted study that examines a novel and important topic. I enjoyed reading it. I have listed some suggestions, by section, below. While I believe you can handle these issues relatively easily, I do think they will strengthen the manuscript and should be addressed. If these issues are addressed, I believe the manuscript warrants publication. Well done!

Introduction

“The most prevalent mechanism responsible for maintaining energy homeostasis during an exercise program is increases in energy intake, largely due to the fact that the rate of energy intake far exceeds the rate of energy expenditure” while I agree with this point, it may be worthwhile to discuss metabolic changes that can occur with weight loss. At least mention adaptive thermogenesis as a possible mechanism complicating weight loss from exercise. It’s possible that alterations in food reinforcement may be part of the adaptive thermogenesis response.

 We certainly agree with this point and have mentioned physiologic adaptations to an exercise-induced energy deficit that also work to return the body back to energy balance, lines 46-48)

Methods

“… 52 completed all baseline tests and were randomized into one of three groups…” I think it’s a good idea to list the three groups parenthetically here. Revie to “… 52 completed all baseline tests and were randomized into one of three groups (six exercise sessions per week, two weekly sessions, sedentary control)…”

 We have made this addition as suggested, line 93.

“The study was a randomized, controlled trial that included a 12-week exercise intervention of either six sessions (days) per week, two sessions per week, or a sedentary control group (no exercise) blocked on gender” list the number of males and females in each group.

 We have included this as suggested on lines 126-127

How was compliance of the sedentary group assessed?

 We have included in our 481-484 that we did not assess the physical activity of the control group during the intervention but believe they remained sedentary as their body weight did not change. We included this in the section on the weaknesses of the study.

What were some of the exercise options available to your exercise groups? Did they only do cardio (e.g., treadmill, bike) or were they able to do resistance training?

 Participants only engaged in aerobic exercise, this is now detailed on lines 135-138

Why was liking only assessed at baseline? Changes in hedonics as a result of the intervention would have been interesting to look at as well as RRV.

 We agree this would have been interesting to include and noted this in the limitations section, lines 462-464.

Interesting uses of Pmax for the RRV task. I like how you quantified RRV but, out of curiosity, why not just use raw Pmax scores for healthy versus unhealthy foods as your measure of RRV?

 We combined the Pmax scores to get an overall food reinforcement score, not relative to anything. But we were also interested if they may increase their reinforcing value of one type of food relative to the other, so maybe they wouldn’t change Pmax of healthy or unhealthy independently, but when put against each other, we may see a significant change in the proportion (maybe people find candy bars more reinforcing than dried banana after exercising). We were worried that assessing the reinforcing value of food relative to a non-food activity (reading or watching TV for example) may have been skewed as people may prefer doing a sedentary activity opposed to exercise if they were burnt out from the intervention. Thus, this would have been more of a test for the reinforcing value of sedentary activities than food and confounded our results. In response to reviewer 1, we provided the results on changes in Pmax for healthy and unhealthy foods in Table 3, the results section, and provided a bit on the discussion on this (lines 409-412)

“Change scores were also tested if significantly different from zero via T-tests.” Were these done for each group separately?

 These were done for each group separately, now indicated on line 263-264..

“Differences in changes in…” revise to “Differences in change scores (post value minus baseline value) in…”

 This has been revised as suggested, lines 476-477

Discussion

“The present findings, demonstrating FFM deficits are the root cause in the increase…” this language is too definitive for my liking. Revise to “The present findings, demonstrating FFM deficits were the root cause in the increase…” or something similar.

 We have revised as suggested, line 439.

“Our findings indicate that there is great variability in individuals’ change in food reinforcement after a 12-week aerobic exercise intervention, and those who do increase their food reinforcement are those who lose the greatest amount of FFM post-intervention…” Again this reads as too definitive. I think it’s important to describe the results only within the context of the present study. Stating “those who do increase their food reinforcement are those who lose the greatest amount of FFM post-intervention” sounds like you are concluding this is what happens with weight loss for everyone. While that may be true, you cannot conclude that from this single study. I know this seems like a small detail, but it is important to remain tentative. Revise to “Our findings indicate that there was great variability in individuals’ change in food reinforcement after a 12-week aerobic exercise intervention, and those who did increase their food reinforcement were also those who lost the greatest amount of FFM post-intervention…” This way you are only talking about your results.

 We agree with this statement and have made the changes to the conclusion as suggested, line 495.

---

## [Decision Letter · Decision Letter 1]

22 May 2020

PONE-D-20-01440R1

The consequences of exercise-induced weight loss on food reinforcement. A randomized controlled trial

PLOS ONE

Dear Dr. Flack,

Thank you for submitting your manuscript to PLOS ONE. After careful consideration, we feel that it has merit but does not fully meet PLOS ONE’s publication criteria as it currently stands. Therefore, we invite you to submit a revised version of the manuscript that addresses the points raised during the review process.

The manuscript was reviewed by a biostatitcian, who pointed several limitation of the manuscript including lack of an *a priori* power assessment and several problems with data analysis. 

We look forward to receiving your revised manuscript.

Kind regards,

Maciej S. Buchowski

Academic Editor

PLOS ONE

Reviewers' comments:

Reviewer's Responses to Questions

**Comments to the Author**

1. If the authors have adequately addressed your comments raised in a previous round of review and you feel that this manuscript is now acceptable for publication, you may indicate that here to bypass the “Comments to the Author” section, enter your conflict of interest statement in the “Confidential to Editor” section, and submit your "Accept" recommendation.

Reviewer #1: All comments have been addressed

Reviewer #3: (No Response)

2. Is the manuscript technically sound, and do the data support the conclusions?

Reviewer #1: Yes

Reviewer #3: Partly

3. Has the statistical analysis been performed appropriately and rigorously? 

Reviewer #1: Yes

Reviewer #3: I Don't Know

4. Have the authors made all data underlying the findings in their manuscript fully available?

Reviewer #1: Yes

Reviewer #3: Yes

5. Is the manuscript presented in an intelligible fashion and written in standard English?

Reviewer #1: Yes

Reviewer #3: Yes

6. Review Comments to the Author

Reviewer #1: My previous revision requests have been addressed. One typo on line 471. Other than that, it is ready for publication.

Reviewer #3: Important note: This review pertains only to ‘statistical aspects’ of the study and so ‘clinical aspects’ [like medical importance, relevance of the study, ‘clinical significance and implication(s)’ of the whole study, etc.] are to be evaluated [should be assessed] separately/independently.

My first question is when title says “A randomized controlled trial” then why this study is classified as [Article Type:] ‘Research Article’? If it is really ‘A randomized controlled trial’ then it is expected to estimate ‘required sample size’. For the study on this topic a small effect size is generally assumed. Even for medium effect size [type I error 5%, Power=80%] according to ‘table-2 on page 158 of an old but classical paper “A power primer” in Psychological Bulletin, 1992, vol.:112, pp 155-159 the required sample size is 52 per group. However, in this study total sample size is only 52. [For small effect size (type I error 5%, Power=80%) it is as large as 322]. Note that in lines 498-500 it is said that “it appears a large sample size is required to better elucidate food reinforcement changes after exercise, potentially taking sex effects into account”. But a priori estimation of the required sample size is not found.

What exactly you want to convey by ‘blocked on gender’ [a sedentary control group (no exercise) blocked on gender], the phrase used (in line 123) while describing group-III? {If that means ‘matched’ why the wide differences (68.4%, 85%, 78.8%, in Gr.I,II,III respectively)? It cannot be usual meaning of ‘block’ because group sizes are so different (19, 20, 14 in Gr.I,II,III respectively)}. Please explain.

According to line 260 “Group differences in baseline study characteristics where tested via 1-way ANOVA” but P-values are not reported in table-2 displayed in lines 505- {which is in fact very good as it is often said that

To provide a description of baseline characteristics is entirely reasonable (since it is clearly important in assessing to whom the results of the trial can be applied), however, it does not require the division of baseline characteristics by treatment groups. Statistical comparison of baseline characteristics is not desirable at all [because even if P-value turns out to be significant (while comparing baseline characteristics despite random allocation), it is, by definition, a false positive] as you then are supposed to be testing ‘randomization’ then, which in any single trial may not balance all baseline characteristics because ‘randomization’ is a sort of ‘insurance’ and not a guarantee scheme}.

If the study was to observe effect of ‘progressive ratio’, then why [as said in lines 148-9] ‘Participants in the two-day per week group were instructed to perform two long exercise sessions per week and encouraged to try to expend 1,000 kcal per session’ is not understood. Will you please thro light on purpose [of that action]. ‘Power Analysis’ account given in lines 271 onwards is not very convincing though ‘was based on our previous study’.

In Table 3 [Resulting data from the 12-week exercise intervention between groups that exercised either six or two days per week and a sedentary control group] data are mean ± SE whereas “It may please be noted that ‘data always should be presented as means ± standard deviation’ [and never as means ± standard errors]. The standard deviation (SD) and standard error (SE) of the mean measures two very different things. The standard error depends heavily upon the sample size. [Note: standard error of the mean tells not about variability in the original population, as the standard deviation does, but about the certainty with which a sample mean estimates the true population mean. Since the certainty with which we can estimate the mean increases as the sample size increases, the standard error of the mean decreases as the sample size increases.] People are interested in knowing variability in ‘study population’ and not precision in estimate.

It is definitely appreciable that [line 294-297] ‘Quantile regression analysis’ [predicting changes in PmaxTotal is presented in Table 5: controlling for weekly exercise energy expenditure, exercise frequency per week (group), and changes in FM, REE and hunger] is used, however, a brief note (at least its non-parametric nature) on this ‘not so popular’ technique was desirable (my opinion may be little bias as a biostatistician). Surprisingly, median or other quartiles are not reported {though this non-parametric technique is used}. As you may be aware, there are disadvantages as well as limitations [like ‘monotonicity constraints’ as pointed out in ‘Journal of Machine Learning Research Non-parametric Quantile Estimation (2005)’ or computations are quite tedious compared to the least squares method] which is why technique, though very old, is seldom used. I believe [not sure] that “quantile regression” (because said in lines 270 that “All analyses were performed) is available in IBM SPSS Version 26.

Limitation mentioned in lines 462-64 [Liking of the test foods was not assessed as baseline liking scores did not influence food reinforcement, but if may have been interesting to see how or if liking of these tests foods changed as a result of the exercise intervention] is questionable. In the backdrop of the clear mention in lines 475-6 that this study is secondary of a larger study, [“the analysis, being a secondary aim of a larger study, was likely underpowered to detect significant differences in change scores”], editors may think about publication of such an article {though other things presented are alright}.

7. PLOS authors have the option to publish the peer review history of their article (what does this mean?). If published, this will include your full peer review and any attached files.

Reviewer #1: No

Reviewer #3: No

---

## [Author Response · Author response to Decision Letter 1]

25 May 2020

Response to Reviewers

We would like to thank the reviewers and Dr. Buchowski for their time and effort into providing their important feedback. We have carefully addressed each comment and feel that this manuscript has improved significantly. Since we satisfied the previous reviewers’ comments, we are responding below only to Reviewer #3’s comments, with our response in italics below each comment.

COMMENTS – Manuscript PONE-D-20-01440R1

Title: “The consequences of exercise-induced weight loss on food reinforcement. A randomized controlled trial”

Important note: This review pertains only to ‘statistical aspects’ of the study and so ‘clinical aspects’ [like medical importance, relevance of the study, ‘clinical significance and implication(s)’ of the whole study, etc.] are to be evaluated [should be assessed] separately/independently.

My first question is when title says “A randomized controlled trial” then why this study is classified as [Article Type:] ‘Research Article’? If it is really ‘A randomized controlled trial’ then it is expected to estimate ‘required sample size’. For the study on this topic a small effect size is generally assumed. Even for medium effect size [type I error 5%, Power=80%] according to ‘table-2 on page 158 of an old but classical paper “A power primer” in Psychological Bulletin, 1992, vol.:112, pp 155-159 the required sample size is 52 per group. However, in this study total sample size is only 52. [For small effect size (type I error 5%, Power=80%) it is as large as 322]. Note that in lines 498-500 it is said that “it appears a large sample size is required to better elucidate food reinforcement changes after exercise, potentially taking sex effects into account”. But a priori estimation of the required sample size is not found. 

Thank you for pointing this out, I initially selected ‘Research Article’ as this is what I believed a manuscript of this nature to fall under, although this study does fit NIH’s description of a clinical trial, so this may need to be updated. I have contacted the journal for advice in changing this designation.

As for the second part of this question, I’m not certain I am understanding this fully. I have provided a power analysis, starting on line 270, and have added detail regarding how and why the study is powered this way. In short, this study is powered to detect a significant difference in body fat loss between groups. Although not specific to the outcome measures of the present investigation, powering the study in this manner allows us the ability to assess the present outcomes in groups that differ in body fat loss, an important, clinically relevant, variable to consider. 

What exactly you want to convey by ‘blocked on gender’ [a sedentary control group (no exercise) blocked on gender], the phrase used (in line 123) while describing group-III? {If that means ‘matched’ why the wide differences (68.4%, 85%, 78.8%, in Gr.I,II,III respectively)? It cannot be usual meaning of ‘block’ because group sizes are so different (19, 20, 14 in Gr.I,II,III respectively)}. Please explain.

You are correct, this wasn’t “blocked” in the traditional sense. Rather, men and women were randomized separately. This has been updated and explained on lines 124/125.

According to line 260 “Group differences in baseline study characteristics where tested via 1-way ANOVA” but P-values are not reported in table-2 displayed in lines 505- {which is in fact very good as it is often said that 

To provide a description of baseline characteristics is entirely reasonable (since it is clearly important in assessing to whom the results of the trial can be applied), however, it does not require the division of baseline characteristics by treatment groups. Statistical comparison of baseline characteristics is not desirable at all [because even if P-value turns out to be significant (while comparing baseline characteristics despite random allocation), it is, by definition, a false positive] as you then are supposed to be testing ‘randomization’ then, which in any single trial may not balance all baseline characteristics because ‘randomization’ is a sort of ‘insurance’ and not a guarantee scheme}.

I do know many statisticians who scoff at the notion of testing for bassline group differences in a randomized trial. Although I’ve also received comments on papers I have submitted where reviewers ask me to test for baseline group differences despite a randomized design. I’m not a statistician, I have no preference, and have therefore removed the mention of testing for group differences (line 260) and simply state that baseline data is presented in table 2 (line 282).

If the study was to observe effect of ‘progressive ratio’, then why [as said in lines 148-9] ‘Participants in the two-day per week group were instructed to perform two long exercise sessions per week and encouraged to try to expend 1,000 kcal per session’ is not understood. Will you please thro light on purpose [of that action]. ‘Power Analysis’ account given in lines 271 onwards is not very convincing though ‘was based on our previous study’. 

The progressive ration (or PR schedules) refers to the food reinforcement assessment. The two and six days of exercise per week were the exercise intervention. We were testing to see if the intervention exerted an effect on the food reinforcement measure (PR schedules). To make this clearer, we have added a couple lines at the end of the I intro, lines 87-89. As noted in the prior comment, we have added detail to the power analysis. 

In Table 3 [Resulting data from the 12-week exercise intervention between groups that exercised either six or two days per week and a sedentary control group] data are mean ± SE whereas “It may please be noted that ‘data always should be presented as means ± standard deviation’ [and never as means ± standard errors]. The standard deviation (SD) and standard error (SE) of the mean measures two very different things. The standard error depends heavily upon the sample size. [Note: standard error of the mean tells not about variability in the original population, as the standard deviation does, but about the certainty with which a sample mean estimates the true population mean. Since the certainty with which we can estimate the mean increases as the sample size increases, the standard error of the mean decreases as the sample size increases.] People are interested in knowing variability in ‘study population’ and not precision in estimate.

We understand this notion of this perspective and the difference in SD and SE, which is why Table 1 included SD, as we are reporting strictly on our sample in this table. I have always been advised to report SE when presenting results from a clinical trial as we are trying to present our data not just specific to our data set but to convey it to the larger population. Although again, if SD is preferred in this scenario we are fine with making these changes. Table 3 has been updated to reflect SD. 

It is definitely appreciable that [line 294-297] ‘Quantile regression analysis’ [predicting changes in PmaxTotal is presented in Table 5: controlling for weekly exercise energy expenditure, exercise frequency per week (group), and changes in FM, REE and hunger] is used, however, a brief note (at least its non-parametric nature) on this ‘not so popular’ technique was desirable (my opinion may be little bias as a biostatistician). Surprisingly, median or other quartiles are not reported {though this non-parametric technique is used}. As you may be aware, there are disadvantages as well as limitations [like ‘monotonicity constraints’ as pointed out in ‘Journal of Machine Learning Research Non-parametric Quantile Estimation (2005)’ or computations are quite tedious compared to the least squares method] which is why technique, though very old, is seldom used. I believe [not sure] that “quantile regression” (because said in lines 270 that “All analyses were performed) is available in IBM SPSS Version 26.

We appreciate the insight, and to better report all quartiles, we have included a histogram for delta PmaxTotal and have included this in the supplemental files and referenced it on line 292.

Limitation mentioned in lines 462-64 [Liking of the test foods was not assessed as baseline liking scores did not influence food reinforcement, but if may have been interesting to see how or if liking of these tests foods changed as a result of the exercise intervention] is questionable. In the backdrop of the clear mention in lines 475-6 that this study is secondary of a larger study, [“the analysis, being a secondary aim of a larger study, was likely underpowered to detect significant differences in change scores”], editors may think about publication of such an article {though other things presented are alright}. 

It is true that this was a secondary aim of a larger study, but the analysis was centered on the regression analysis, identifying predictors in food reinforcement. Just because there were no statistically significant changes in Pmax, the regression analysis was still able to demonstrate significant findings. With changes in PmaxTotal over +20 in the 6-d group and -39 and -14 in the 2-d and control groups, respectively, this seems like a fair speculation and worthy of publication, albeit probably not in Nature or N Engl J Med. This notion that liking may have changed during the intervention to influence food reinforcement at post-testing was brought up by another reviewer before, questioning if liking was assessed at 12-weeks and if there were changes in this measure. Since we did not assess liking at 12 weeks, we considered this a limitation.

---

## [Decision Letter · Decision Letter 2]

2 Jun 2020

The consequences of exercise-induced weight loss on food reinforcement. A randomized controlled trial

PONE-D-20-01440R2

Dear Dr. Flack,

We are pleased to inform you that your manuscript has been judged scientifically suitable for publication and will be formally accepted for publication once it complies with all outstanding technical requirements.

With kind regards,

Maciej S. Buchowski

Academic Editor

PLOS ONE

Additional Editor Comments (optional):

Reviewers' comments:

Reviewer's Responses to Questions

**Comments to the Author**

1. If the authors have adequately addressed your comments raised in a previous round of review and you feel that this manuscript is now acceptable for publication, you may indicate that here to bypass the “Comments to the Author” section, enter your conflict of interest statement in the “Confidential to Editor” section, and submit your "Accept" recommendation.

Reviewer #3: (No Response)

2. Is the manuscript technically sound, and do the data support the conclusions?

Reviewer #3: (No Response)

3. Has the statistical analysis been performed appropriately and rigorously? 

Reviewer #3: (No Response)

4. Have the authors made all data underlying the findings in their manuscript fully available?

Reviewer #3: (No Response)

5. Is the manuscript presented in an intelligible fashion and written in standard English?

Reviewer #3: (No Response)

6. Review Comments to the Author

Reviewer #3: (No Response)

7. PLOS authors have the option to publish the peer review history of their article (what does this mean?). If published, this will include your full peer review and any attached files.

Reviewer #3: No

---

## [Editor Report · Acceptance letter]

8 Jun 2020

PONE-D-20-01440R2 

The consequences of exercise-induced weight loss on food reinforcement. A randomized controlled trial 

Dear Dr. Flack:

I'm pleased to inform you that your manuscript has been deemed suitable for publication in PLOS ONE. Congratulations! Your manuscript is now with our production department. 

Kind regards, 

on behalf of

Dr. Maciej S. Buchowski 

Academic Editor

PLOS ONE